# Differential Analysis of Island Mountain Plant Community Characteristics: Ecological Sensitivity Perspectives

**Jinyan Liu [1,2,†], Junyi Li [1,†], Daoyuan Chen [1], Linye Guo [1], Guochang Ding [1,*] and Jianwen Dong [1,*]**

[1] College of Landscape Architecture and Art, Fujian Agriculture and Forestry University, 63 Xiyuangong Rd., Fuzhou 350002, China; fjljy@qztc.edu.cn (J.L.); 22319075007@fafu.edu.cn (J.L.); 12319075054@fafu.edu.cn (D.C.); 12319075047@fafu.edu.cn (L.G.)

[2] College of Resource and Environmental Sciences, Quanzhou Normal University, 398 Donghai Rd., Quanzhou 362000, China

[*] Correspondence: fjdgc@fafu.edu.cn (G.D.); fjdjw@fafu.edu.cn (J.D.);
Tel.: +86-132-3590-5585 (G.D.); +86-136-0952-5156 (J.D.)

[†] These authors contributed equally to this work.

**Abstract:** Island plants form the foundation for maintaining the ecology of an island. With the development of the island's infrastructure, its ecosystems become damaged to a certain extent. A comprehensive understanding of island habitats and plant community characteristics is crucial for the development of island plant communities. This paper focuses on Pingtan Island in Fujian Province, China, as the research subject. Firstly, considering the significance of the wind environment on the island, this study constructed a wind environment model for the entire island of Pingtan to evaluate the ecological sensitivity from a macro perspective. Subsequently, 33 typical sample plots were selected based on different ecologically sensitive areas to conduct a micro-survey and the characterization of the montane plant communities on Pingtan Island. The findings reveal that (1) Pingtan Island's ecological sensitivity is dominated by areas with ecological insensitivity (35.72%), moderate ecological sensitivity (33.99%), and high ecological sensitivity (18.02%). The soil texture, wind environment, and land use type are the primary influencing factors in the ecological sensitivity of Pingtan Island. (2) A total of 47 families, 82 genera, and 93 species of plants were investigated in a typical sample site in the mountainous area of Pingtan Island. The plant community structure was dominated by the successional stage of shrubs and herbs. There is some similarity in the plant composition of different ecologically sensitive areas. High ecologically sensitive areas have more species. As sensitivity increases, the dominant species in the three ecologically sensitive areas continue to undergo plant succession from *Acacia confusa* to *Pinus thunbergii* to *Eurya emarginata*. (3) Both community characteristics and species diversity vary between sensitive areas. The canopy density (CD) and the mean height of tree layer (MHTL) are higher in moderate ecologically sensitive areas. The mean tree diameter at breast height (MDBH) and the mean height of shrub layer (MHSL) are higher in high ecologically sensitive areas, while the mean height of herb layer (MHHL) is higher in extreme ecologically sensitive areas. Four diversity indicators increase with increasing sensitivity. In the moderate and high ecologically sensitive areas, *Casuarina equisetifolia* and *A. confusa* thrive, with *Pinus thunbergii* showing the opposite trend. However, species diversity is better characterized by *A. confusa* and *P. thunbergii*, with *C. equisetifolia* being the least diverse. Both the community characteristics and species diversity of *P. thunbergii* are optimal in extreme ecologically sensitive areas. In this study, the ecological sensitivity of Pingtan Island and the characteristics of montane plant communities were systematically analyzed to explore more stable montane plant communities on the island, aiming to provide a scientific basis and model reference for the ecological restoration and sustainable development of Pingtan Island and other islands.

**Keywords:** Pingtan Island; wind environment simulation; plant community composition; species diversity

## 1. Introduction

With the rapid advancement of the globalized economy, nations worldwide have prioritized marine development as a key strategic objective. Serving as the forefront in marine economic development [1], islands possess immeasurable political, economic, military, scientific research, and ecological significance. Nevertheless, islands, being quintessential fragile ecosystems, commonly grapple with ecological and environmental challenges, such as the scarcity of freshwater resources, infertile soil, and limited biodiversity [2,3]. The 908 specialized studies focused on island coastal zones completed post-2011 reveal that global warming, the surge in extreme weather events, and prolonged, unregulated, and crude large-scale development have induced alterations in island topography and geomorphology, thereby exacerbating the degradation of island ecosystems [3]. In contrast to mainland areas, island ecosystems face formidable challenges in recovering once disturbed or damaged and may even undergo irreversible and accelerated deterioration [4]. Consequently, the protection of islands has evolved into a global ecological imperative that demands unwavering attention. Presently, within the framework of addressing challenges related to biodiversity conservation and ecological security, China is dedicating significant efforts to the protection, restoration, and sustainable tourism of its islands.

Ecological sensitivity pertains to the degree of responsiveness of ecosystems to internal and external pressures, such as changes in the natural environment and disturbances from human activities, within a given region [5]. Conducting ecological sensitivity assessments to identify potential ecological challenges can furnish essential foundational information for addressing environmental issues. The construction of a suitable evaluation system, tailored to the characteristics of the study area, stands as a particularly pivotal aspect of an ecological sensitivity assessment. In the extant literature, factors such as elevation, soil composition, water quality, and vegetation coverage are commonly considered in ecological sensitivity evaluations [5–7]. It is noteworthy that the habitat characteristics of sea islands, owing to their unique geographical locations, can be significantly influenced by the wind environment. Strong winds, for instance, may carry a substantial amount of sand, accelerating soil erosion and rendering many plants unable to thrive or grow optimally. This, in turn, results in the progressive deterioration of environmental quality. However, the incorporation of this evaluation metric is not widespread in current research [8]. This is due to the complexity of the island wind environment, which is both seasonal and stochastic, posing challenges in obtaining accurate research data. Commonly employed methods for data collection in wind environment studies encompass field tests, wind tunnel tests, and numerical simulation methods [9–11]. In comparison, numerical simulation is favored for its efficiency in terms of time and cost savings. Computational Fluid Dynamics (CFD) simulation, as a common method in numerical simulation, is widely used at the micro- and mesoscales for simulating wind environments, such as building monoliths and communities [12,13]. The method is considered to be able to better handle complex urban downgradient surfaces, allowing a refined representation of the internal wind field of the modeled object, and has proved to be reliable and superior. However, due to the high dependence on computer resources, research applications of CFD simulations at larger scales are still relatively rare at present. Based on this, this paper proposes to introduce the CFD simulation method to establish the wind environment model of Pingtan Island to determine the construction method of a large-scale wind environment model and the selection of suitable simulation software. Wind environment data were extracted as a crucial indicator for ecological sensitivity evaluation, and the evaluation system was refined to provide more detailed data support for the habitat assessment of Pingtan Island.

The island's montane plants, as the island's most important green patches, form the skeleton of the island's main ecological patches. Typical plants in the mountainous areas of the island are usually common and suitable plants or zonal plants of mountainous areas. After a long period of reproduction, renewal, and succession, they became the dominant species with certain ecological niches, wider distributions, and more quantitative characteristics on the island. At present, plant restoration through artificial forest planting

and ecological protection policy management and other human-regulated methods has become a common ecological restoration method for islands. However, due to the specificity and sensitivity of the environment, there are few species of plants suitable for growing in the island mountains; the biodiversity is extremely low, and there is often poor growth, which varies from one sensitive area to another. On the other hand, due to the lack of sufficient basic information and technical means, a plant used for ecological restoration on islands is often eliminated because it is not adapted to the environment [14]. Based on this, do the differences among the different ecologically sensitive areas of the islands lead to different characteristics of their montane plant communities? What are the characteristics of relatively stable montane plant communities in different ecologically sensitive areas? This will help to enrich the basic information on mountain plants on the islands and will be of great significance in guiding the targeted protection of island ecosystems and the promotion of plant restoration.

Pingtan Island, the largest island in Fujian Province, China, serves as a crucial ecological barrier along the southeast coast of the country. Positioned as a significant gateway for exchanges between mainland China and Taiwan Province, Pingtan Island holds strategic importance for the nation's economy and ecology. Recognized as a typical ecologically fragile island region [15], Pingtan Island is confronted with increasingly prominent ecological security challenges amidst rapid urbanization, mirroring the issues faced by other islands [16] Moreover, being one of the high-wind regions in the country [17], Pingtan Island is consistently impacted by wind and sand throughout the year. Previous studies have indicated that, in Pingtan, the closer one is to the coastline and the higher the elevation, the greater the wind speed, resulting in diminished ecological stability and a deteriorating environment [18]. This circumstance further exacerbates reforestation efforts, posing exceptional challenges [19]. It is well known that plants play an important role in regulating and improving ecological environment management, such as windbreak and sand fixation. As the important green ecological skeleton of Pingtan Island, mountain plants are very important to the ecological environment of Pingtan Island. Pingtan Island belongs to the subtropical evergreen broad-leaved forest vegetation zone. The primary vegetation has been mostly destroyed and is now dominated by secondary vegetation, supplemented by artificial protection forests. The main dominant species include *Pinus thunbergii*, *Acacia confusa*, and *Casuarina equisetifolia* [20]. Ecological succession generally progresses through four successional stages: scrub, coniferous forest, coniferous and broad-leaved mixed forest, and evergreen broad-leaved forest [21]. In 2009, following a change in administrative divisions to establish an independent administrative status, Pingtan Island underwent significant landscape renovations. These alterations, whether intentional or unintentional, introduced exotic plants and potentially heightened the island's ecological vulnerability [15,22,23]. Pingtan Island encompasses a botanical diversity comprising 541 species representing 127 families and 369 genera of seed plants. Among these, 330 species are indigenous, including exemplars of typical island native flora, such as *Eurya emarginata* and *Ipomoea pes-caprae*. Additionally, there are 211 exotic plant species, with some naturalized plants, including *Oenothera drummondii*, *Ipomoea cairica*, *Pinus thunbergii*, *Casuarina equisetifolia*, *Acacia auriculiformis*, and *Acacia confusa* [20]. Given these considerations, selecting Pingtan Island as the focal point for this study becomes both typical and imperative. This study aims to conduct an evaluation of the island's ecological sensitivity and to investigate and characterize mountain plants. The objectives of this work include the following:

(1) Conduct a comprehensive evaluation of the ecological sensitivity of Pingtan Island as a whole, with a systematic assessment of its ecological sensitivity characteristics;

(2) Undertake a survey of plant resources in the mountains of Pingtan Island to identify the composition of species and dominant species in different ecologically sensitive areas. Additionally, the study aims to explore the characteristics and species diversity of plant communities across various ecologically sensitive areas.

This study aims to identify a relatively stable model for plant communities in the mountainous region of Pingtan Island. The findings hold significant implications for the

ecological restoration and landscape optimization of Pingtan Island. Furthermore, this study serves as a valuable reference for landscape construction on other islands. The flowchart of the study is shown in Figure 1.

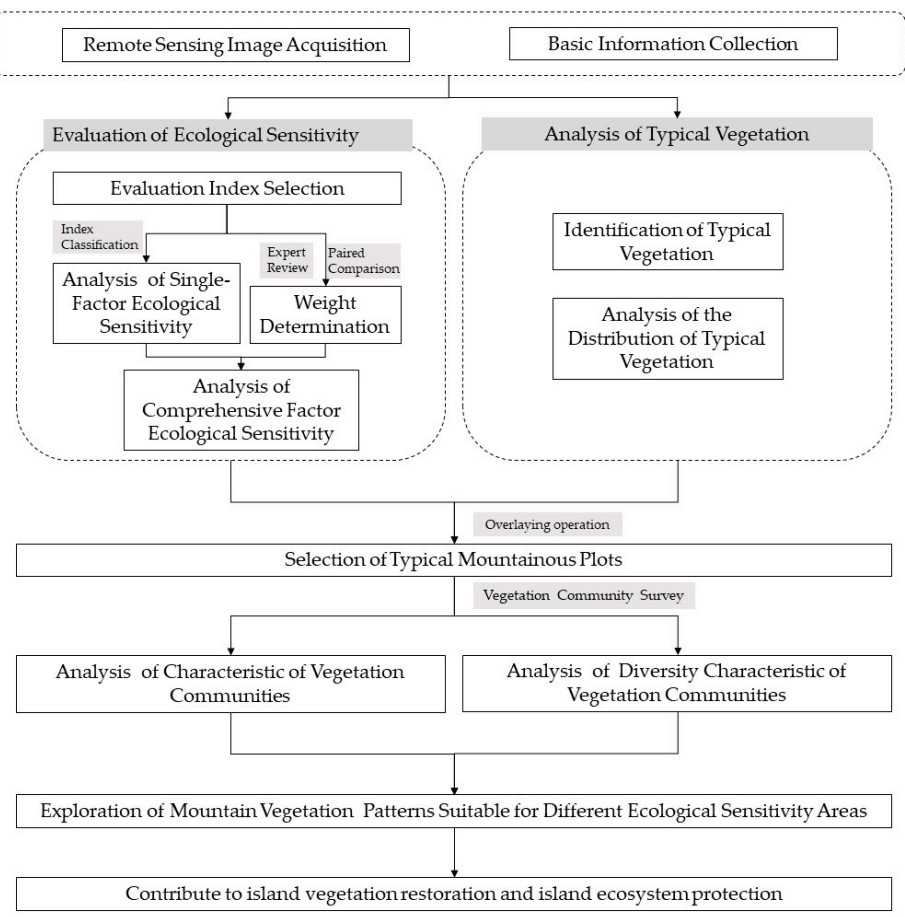

**Figure 1.** Flowchart of the study.

## 2. Materials and Methods

### 2.1. Study Area

#### 2.1.1. Location and Climate

Pingtan Island is situated in the eastern sea of China, at a latitude range of $25°15'$ to $25°45'$ N and a longitude range of $119°32'$ to $120°10'$ E (Figure 2). It borders the Taiwan Strait to the east and is separated by the Haitan Strait to the west. Pingtan Island experiences a southern subtropical semi-moist marine monsoon climate, characterized by an average annual temperature ranging from 19.0 to 19.9 °C and average annual precipitation between 900 and 1200 mm. The island lacks rivers but features 46 seasonal creeks [24]. The average annual wind speed is 6.9 m/s, with 125 days experiencing gale force 7 or above throughout the year. Northeasterly winds prevail, except in summer, when southwesterly winds are prevalent [16].

#### 2.1.2. Topography and Geomorphology

Pingtan Island features elevated terrain in the north and south, characterized by undulating hills and low mountains, with the central region exhibiting lower elevation, primarily comprising a marine plain topography [2]. The mountainous and hilly areas are primarily divided into three sections, namely, northeast, northwest, and southern regions, covering a total area of 159 km$^2$, representing 42% of the total land area. The elevations in the northeastern mountainous region are mostly above 200 m, ranging between 100 and 180 m in the northwestern mountainous region and varying from 100 to 250 m above

sea level in the southern mountainous region. Pingtan Island is located at the northeast terminus of the Changle–Zhao'an active fault belt, and the rock outcrop is mainly of igneous origin from the late Yanshanian. Approximately 66% of the total island area is covered by aeolian sand, contributing to the formation of a typical coastal aeolian geomorphic landscape in southern China [25].

### 2.1.3. Soils and Plants

Pingtan soils have a total of 6 soil classes, 25 soil genera, and 34 soil species, which are commonly characterized by thin soil layers, low nutrient content, and a more obvious vertical distribution of zonal soils [26]. The island's soil is dominated by brick-red loamy red soil (46%), coastal aeolian sandy soil (23.3%), and saline soil [27], followed by paddy soil, red soil, and alluvial soil. The woodland soils belong to three soil classes: brick-red loamy red soil, coastal aeolian sandy soil, and saline soil, with a total of five subclasses and eight soil genera. The soils in the main plant communities are all acidic, and the closer they are to the coastline, the less acidic they are [28]. The soil pH value, soil nitrogen content, organic matter content, available potassium content, and altitude have important effects on the distribution of communities [29].

Afforestation and greening efforts have significantly increased Pingtan's forest coverage, rising from 0.3% in 1949 to 29% in 2008 and reaching 37.2% in 2019. The greening coverage of the built-up area has reached 40.84% (2019). However, due to natural constraints, the construction of ecological landscape forests on Pingtan Island still faces challenges, resulting in poor silvicultural conditions [23]. The predominant plant species include island protection forest species, notably *Acacia confusa*, *Pinus thunbergii*, *Casuarina equisetifolia*, and *Pinus elliottii*, as well as some ornamental garden plants.

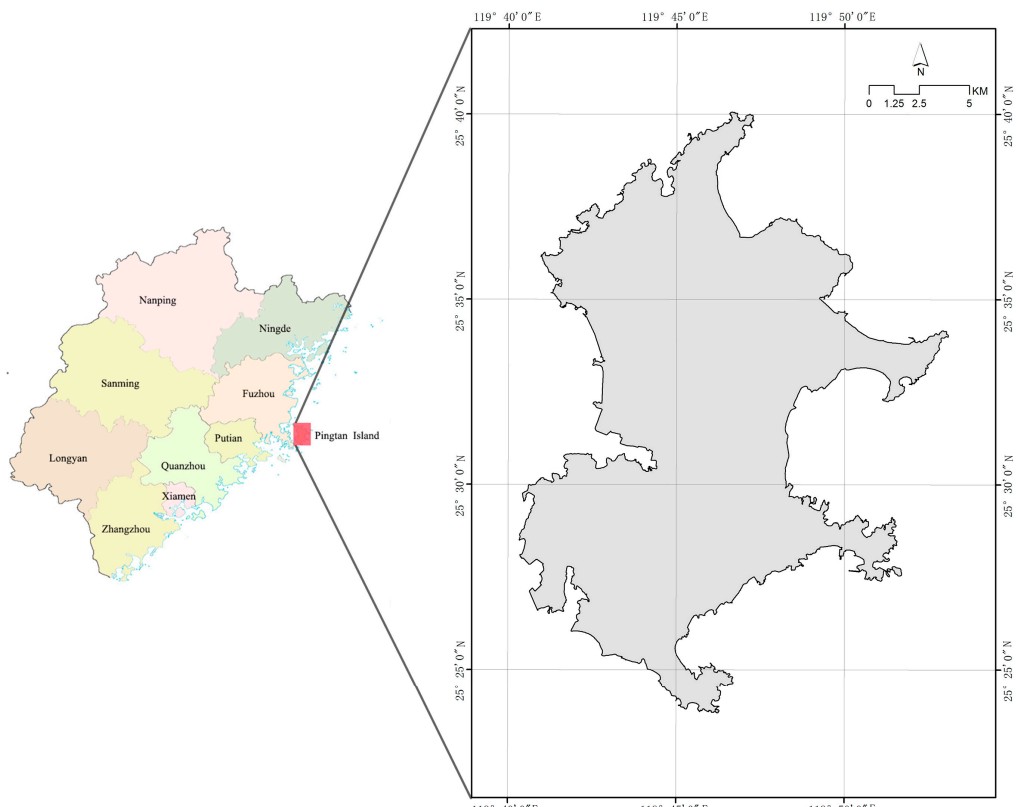

**Figure 2.** Location of study area.

## 2.2. Datasets

The data for this study were sourced from the geospatial data cloud (http://www.gscloud.cn/ accessed on 10 December 2023), encompassing the 2020 Landsat OLI_TIRS remote sensing image of Pingtan Island with a resolution of 30 m × 30 m. Additionally, the dataset included the 30 m Pingtan Data Elevation Model (DEM), the 2020 Forestry Second Survey data of Pingtan, the Pingtan County Forest Vegetation Survey Report (1985), and meteorological data spanning the past 50 years in Pingtan (retrieved from the China Meteorological Data Network), among other sources.

## 2.3. Methods

### 2.3.1. Evaluation of Ecological Sensitivity

#### Selection of Ecological Sensitivity Indicators

Given the prevalence of the northeast wind on Pingtan Island from October to April of the succeeding year and the island's maximum elevation reaching 438.7 m, factors such as elevation, slope, and slope direction exert a considerable influence on ecological elements, including plant communities, soil, and water [30,31]. Among these factors, wind significantly affects plants. Due to challenges in obtaining measured data, the wind environment is simulated for an ideal terrain [13], although adjustments are made based on actual observations. The impact of anthropogenic interference on the ecological environment is predominantly evident in the land use type. Drawing on previous studies [5,7,32] and taking into account the ecological conditions of Pingtan Island, a set of ecological sensitivity evaluation factors for Pingtan Island was initially formulated, and expert consultation was conducted. Ultimately, eight factors, namely, elevation, slope, slope direction, vegetation coverage, soil texture, water environment, wind environment, and land use type, were selected to construct the ecological sensitivity evaluation system for Pingtan Island.

#### Quantitative Decomposition of Ecological Sensitivity Indicators

Quantitative data for each factor, except for the wind environment, were obtained through remote sensing imagery, second forestry survey data, and field research. Given the complexity of the wind environment on Pingtan Island and the extensive research scope, this study employed Computational Fluid Dynamics (CFD) software, such as Phoenics 2019, to simulate the wind environment. This involved combining meteorological data from the past 50 years on Pingtan Island. The specific processes included the following steps: (1) Contour extraction. Contour lines were extracted using ArcGIS 10.5 software. (2) Pre-experimentation with local and regional wind environment simulations. The model of Wangye Mountain was established in AutoCAD and SketchUp and then simulated in Phoenics to determine the most suitable simulation software. (3) Contour thinning and modeling. Based on the standard of setting the contour below 100 m above sea level to 10 m and the contour above 100 m to 20 m for thinning treatment, SketchUp2019 sandbox tools were used to generate the contour model of Pingtan Island. (4) Wind field grid setting. The grid density was gradually increased from both ends to the middle through several simulation iterations to minimize errors generated by wind speed decay during the calculation. (5) Parameter setting. Parameters such as NNE wind direction, 13.9 m/s wind speed, and 1.5 m height near the ground were adopted to simulate the wind environment.

#### Grading of Ecological Sensitivity Indicators

Factor assignment is based on the "principle of largeness" and the ecological "principle of least restriction". The evaluation indicators are categorized into 5 levels based on the degree of importance, with each factor level assigned a uniform score to facilitate horizontal comparisons. The final ecological sensitivity evaluation system established for Pingtan Island is presented in Table 1.

**Table 1.** Ecological sensitivity evaluation of Pingtan Island.

| Target Layer | Indicator Layer | Factor Layer | Criteria for Assigning Points | Degree Value | Degree of Ecological Sensitivity |
|---|---|---|---|---|---|
| Ecological Sensitivity Evaluation of Pingtan Island | Natural Factors | Elevation | 0–50 m | 1 | Ecological insensitivity |
| | | | 50–100 m | 3 | Slight ecological sensitivity |
| | | | 100–150 m | 5 | Moderate ecological sensitivity |
| | | | 150–250 m | 7 | High ecological sensitivity |
| | | | ≥250 m | 9 | Extreme ecological sensitivity |
| | | Slope | ≤5° | 1 | Ecological insensitivity |
| | | | 5–15° | 3 | Slight ecological sensitivity |
| | | | 15–25° | 5 | Moderate ecological sensitivity |
| | | | 25–35° | 7 | High ecological sensitivity |
| | | | ≥35° | 9 | Extreme ecological sensitivity |
| | | Slope direction | South, Flat Slope | 1 | Ecological insensitivity |
| | | | Southwest, southeast | 3 | Slight ecological sensitivity |
| | | | West | 5 | Moderate ecological sensitivity |
| | | | Northwest, east | 7 | High ecological sensitivity |
| | | | North, northeast | 9 | Extreme ecological sensitivity |
| | | Vegetation coverage | 0–0.2 | 1 | Ecological insensitivity |
| | | | 0.2–0.4 | 3 | Slight ecological sensitivity |
| | | | 0.4–0.6 | 5 | Moderate ecological sensitivity |
| | | | 0.6–0.8 | 7 | High ecological sensitivity |
| | | | >0.8 | 9 | Extreme ecological sensitivity |
| | | Soil texture | Others | 1 | Ecological insensitivity |
| | | | Brick-red loamy red soil | 3 | Slight ecological sensitivity |
| | | | Lateritic red soil | 5 | Moderate ecological sensitivity |
| | | | Red soil | 7 | High ecological sensitivity |
| | | | Coastal aeolian sandy soil and saline soil | 9 | Extreme ecological sensitivity |
| | | Water environment | >500 m buffers | 1 | Ecological insensitivity |
| | | | 200–500 m buffers | 3 | Slight ecological sensitivity |
| | | | 100–200 m buffers | 5 | Moderate ecological sensitivity |
| | | | 50–100 m buffers | 7 | High ecological sensitivity |
| | | | Reservoirs and lakes and within their 50 m buffers | 9 | Extreme ecological sensitivity |
| | | Wind environment | 0–9 m/s | 1 | Ecological insensitivity |
| | | | 9–15 m/s | 3 | Slight ecological sensitivity |
| | | | 15–18 m/s | 5 | Moderate ecological sensitivity |
| | | | 18–27 m/s | 7 | High ecological sensitivity |
| | | | 27–48 m/s | 9 | Extreme ecological sensitivity |
| | Human Factors | Land use type | Building land | 1 | Ecological insensitivity |
| | | | Bare ground | 3 | Slight ecological sensitivity |
| | | | Farmland | 5 | Moderate ecological sensitivity |
| | | | Woodland, grassland | 7 | High ecological sensitivity |
| | | | Water | 9 | Extreme ecological sensitivity |

Determination of Ecological Sensitivity Evaluation Indicator Weights

A paired comparison test and the expert consultation method were used to determine the weights of each evaluation factor. The results indicated the following weights for each factor: elevation 0.118, slope 0.074, slope direction 0.094, vegetation coverage 0.136, soil texture 0.193, water environment 0.081, wind environment 0.162, and land use type 0.143.

Spatial Overlay Analysis of Ecological Sensitivity of Composite Factors

The spatial analysis function of ArcGIS was employed to overlay the 8 factor indices and their corresponding weight values. This process calculated the comprehensive index of ecological sensitivity for Pingtan Island and reclassified it into a 5-level comprehensive ecologically sensitive area.

2.3.2. Sample Plot Selection

Determination of Typical Plant Communities on Pingtan Island

Utilizing data from the second forestry survey, the main plant communities on Pingtan Island were organized with the assistance of ArcGIS, providing a reference basis for subsequent sample plot selection. As presented in Table 2, the main plant communities on Pingtan Island consist of five species, with *Acacia confusa* plant communities covering a total area of 38.526% of the island. The research findings suggested that *Eucalyptus robusta* plant communities have limited landscape ecological benefits. Therefore, this study opted to focus on the four most typical plant communities—*Pinus thunbergii*, *Pinus elliottii*, *Acacia confusa*, and *Casuarina equisetifolia*—for further investigation.

**Table 2.** Area of main plant communities in Pingtan.

| No. | Plant Community | Size (km$^2$) | Percentage (%) |
|---|---|---|---|
| 1 | *Pinus thunbergii* | 3.782 | 1.199 |
| 2 | *Pinus elliottii* | 1.617 | 0.513 |
| 3 | *Eucalyptus robusta* | 2.965 | 0.940 |
| 4 | *Acacia confusa* | 67.210 | 21.307 |
| 5 | *Casuarina equisetifolia* | 45.952 | 14.567 |

Typical Sample Site Selection Based on Typical Plants and Ecological Sensitivity

The survey identified that the aforementioned four typical plant communities are more extensively distributed in Jun Mountain, Wangye Mountain, and Niuzhai Mountain, displaying a certain degree of representativity. Simultaneously, an overlay analysis revealed that the mountains on Pingtan Island are predominantly situated in moderate, high, and extreme ecologically sensitive areas. Consequently, this study designated these three mountains as the primary research areas and ultimately selected typical sample plots based on the three types of ecologically sensitive areas for investigation. In the end, 33 research plots were determined, as depicted in Figure 3.

Plant Community Survey Methods

The investigation employed typical plant community survey methods for basic information gathering [33]. Tree-related information, such as names, growth conditions, canopy density, diameter at breast height, height, and other details, were recorded using a circumference ruler and Leica laser rangefinder D81 (LeicaDISTOD5, Heerbrugg, Switzerland) for trees (diameter at breast height $\geq$ 3 cm or tree height $\geq$ 3 m) and other species in the plot [34]. In addition, small quadrats were randomly selected from each forest quadrat to measure shrub layer plants (5 m $\times$ 5 m) and ground cover plants (1 m $\times$ 1 m). Information, including the number, height, crown width, coverage, name, and other relevant details of shrubs and herbs, was documented.

Plant communities were classified based on plant community ecological principles [33]. The top five plant species with significant values were determined by calculating the importance values, leading to the identification of communities dominated by dominant species.

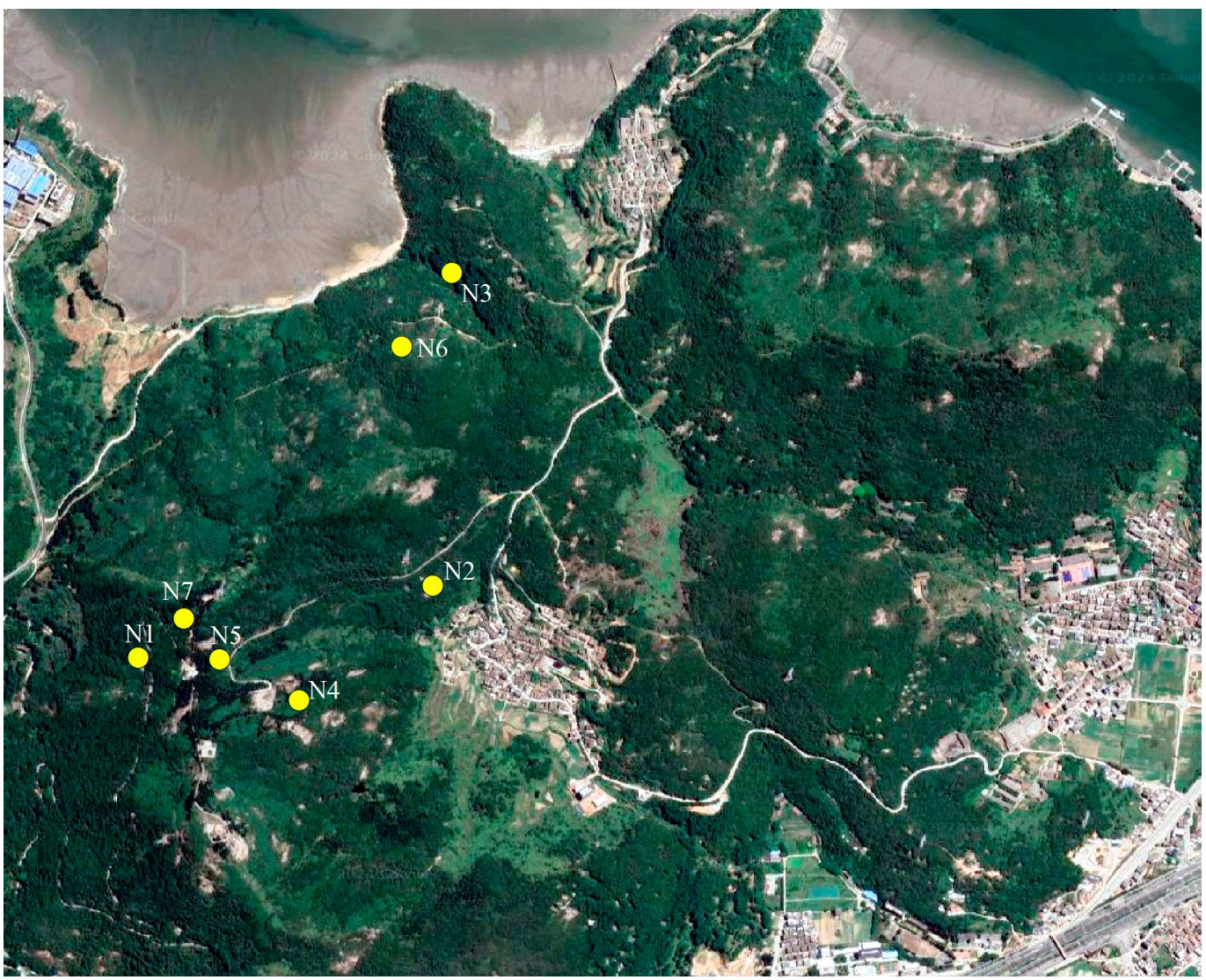

(**a**) Niuzhai Mountain

**Figure 3.** *Cont.*

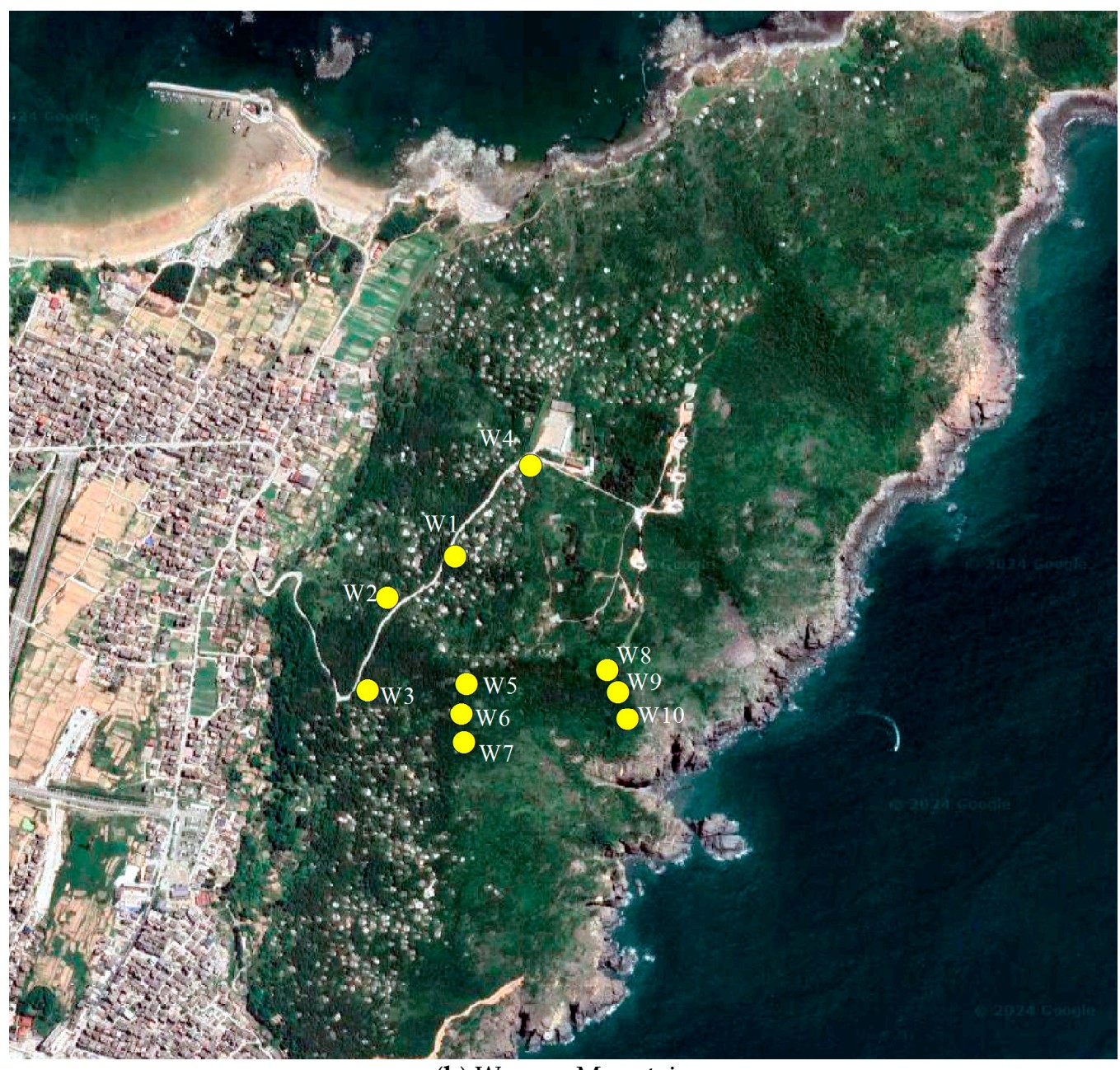

(**b**) Wangye Mountain

**Figure 3.** *Cont.*

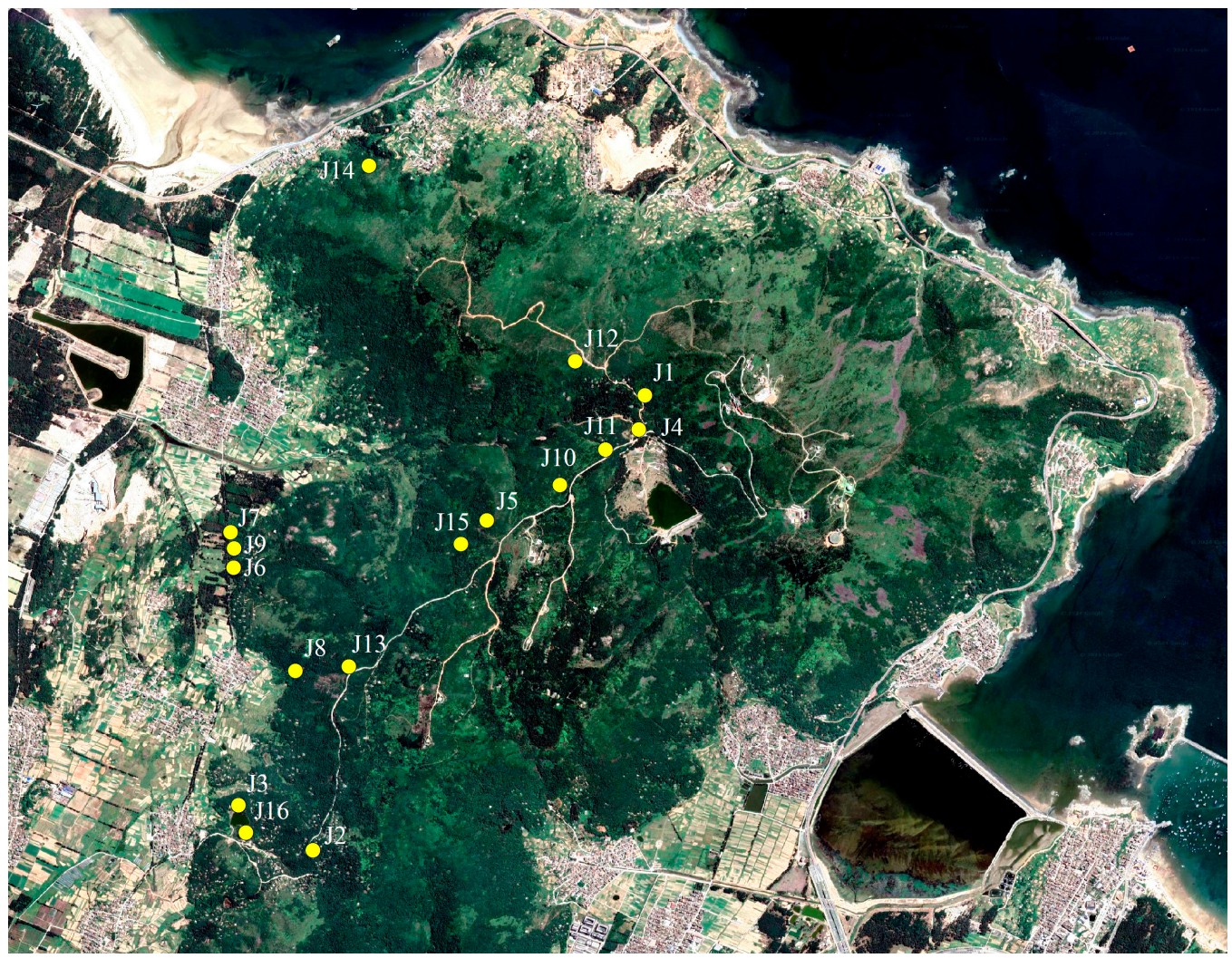

(**c**) Jun Mountain

**Figure 3.** Typical plot distribution map of Pingtan Island.

2.3.3. Geostatistical Analysis

The data analysis in this study encompassed the determination of species importance values and species diversity. The species importance value determination method was utilized to measure the trees, shrubs, and grass layers individually. The comprehensive importance value was obtained. Owing to the straightforward structural composition of the surveyed island plant communities and the conspicuous dominance of certain species, we designated the dominant species as the foundational components for community establishment. The classification of a plant community was determined based on one or more species with the highest importance value [35], and the group average clustering method was employed to classify plant communities. Species diversity was measured according to the method outlined by Ma et al. [36], incorporating indicators such as the Margalef index, Shannon–Wiener index, Simpson index, and Pielou index.

## 3. Results

### *3.1. Analysis of Ecological Sensitivity Evaluation of Pingtan Island*

3.1.1. Single-Factor Ecological Sensitivity Analysis

Figure 4 illustrates the ecological sensitivity characteristics of various factors. Both the elevation and slope of Pingtan Island are predominantly characterized by insensitivity and slight sensitivity, accounting for 93.03% and 85.65%, respectively, indicating a relatively gentle overall terrain. The slope orientation is evenly distributed, with high sensitivity (24.15%) and slight sensitivity (24.11%) dominating, particularly for northwest and east slope orientations, as well as southwest and southeast slope orientations. Vegetation coverage is mainly characterized by extreme sensitivity (28.23%) and high sensitivity (23.54%), primarily located in typical mountain areas, some lower mountains, and forest belts, with a lower percentage in urban built-up areas and certain coastal regions, signifying favorable vegetation coverage on Pingtan Island. The soil is mostly insensitive (53.19%), prevalently in cultivated land and urban built-up areas. This is followed by moderate and extreme sensitivities (21.96% and 21.46%, respectively), distributed in areas below 200 m in elevation, in the locations of major windbreaks, and near the western coast. The water environment is dominated by insensitivity (77.48%). Due to topographical constraints, the surface water on Pingtan Island is limited, exerting a minor impact on island habitats. Wind environment sensitivity is mainly moderate (42.32%) and slight (41.44%), corresponding to wind speeds primarily between 15 and 18 m/s and between 9 and 15 m/s, with extreme sensitivity (0.64%) concentrated in the five major vents. The land use type shows predominantly moderate sensitivity (48.34%) and high sensitivity (27.91%), encompassing arable land, woodland, and grassland, while extreme sensitivity (1.72%) is minimal. Despite being relatively water-scarce, the water areas are the most fragile, prone to drying up and disappearing.

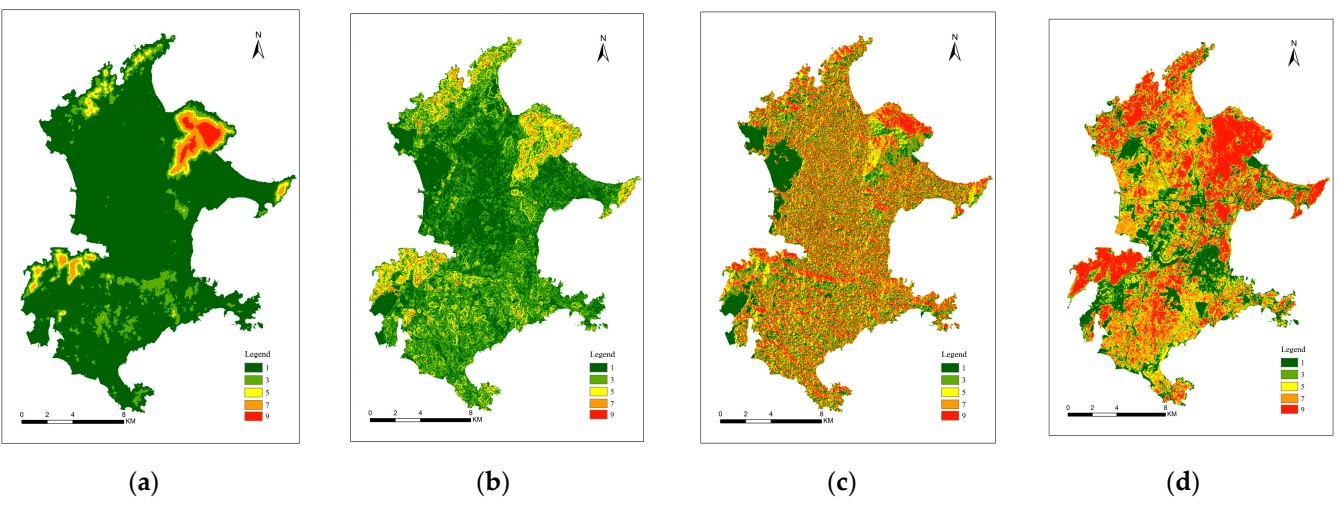

| (**a**) | (**b**) | (**c**) | (**d**) |

**Figure 4.** *Cont*.

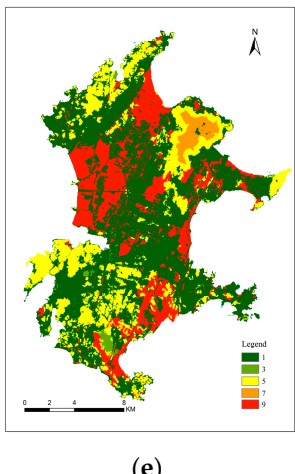 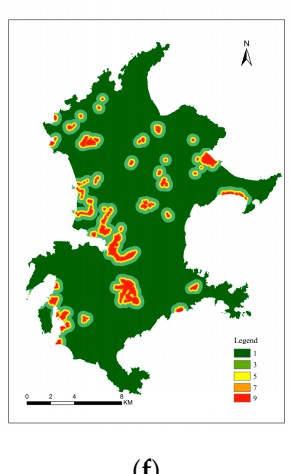 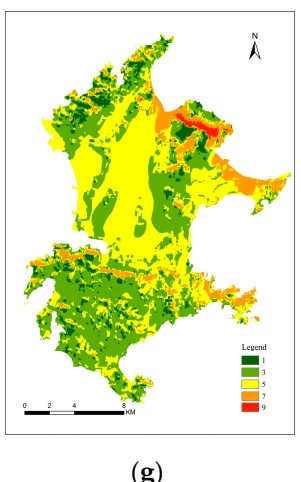 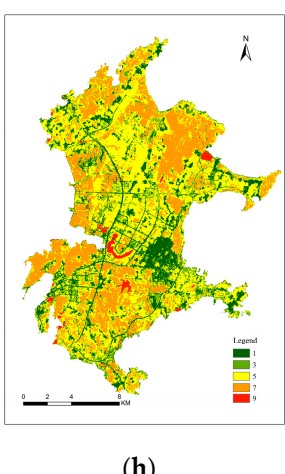

(**e**) (**f**) (**g**) (**h**)

**Figure 4.** Single-factor ecological sensitivity analysis maps of Pingtan Island. (**a**) Elevation Grading Map, (**b**) Slope Grading Map, (**c**) Slope Direction Grading Map, (**d**) Vegetation Coverage Grading Map, (**e**) Soil Texture Grading Map, (**f**) Water Environment Grading Map, (**g**) Wind Environment Grading Map, (**h**) Land Use Type Grading Map. Note: 1 represents ecological insensitivity; 3 represents slight ecological sensitivity; 5 represents moderate ecological sensitivity; 7 represents high ecological sensitivity; 9 represents extreme ecological sensitivity.

3.1.2. Comprehensive Factor Ecological Sensitivity Analysis

As shown in Table 3 and Figure 5, the comprehensive ecological sensitivity of Pingtan Island is presented as follows: slightly ecologically sensitive area (35.72%) > moderate ecologically sensitive area (33.99%) > high ecologically sensitive area (18.02%) > ecologically insensitive area (10.79%) > extreme ecologically sensitive area (1.48%).

**Table 3.** Comprehensive ecological sensitivity analysis of Pingtan Island.

| No. | Degree of Ecological Sensitivity | Size (km$^2$) | Percentage (%) |
|-----|----------------------------------|---------------|----------------|
| 1 | Ecological insensitivity | 29.65 | 10.79 |
| 2 | Slight ecological sensitivity | 98.18 | 35.72 |
| 3 | Moderate ecological sensitivity | 93.41 | 33.99 |
| 4 | High ecological sensitivity | 49.53 | 18.02 |
| 5 | Extreme ecological sensitivity | 4.07 | 1.48 |

The slightly ecologically sensitive areas are primarily situated in some of the lower hilly areas in the central and western parts of the island, interspersed with slightly sensitive areas. The moderate ecologically sensitive areas include low mountainous rural settlements around the central 36-foot lake and the low-slope areas of slightly higher mountains, such as Jun Mountain and Wangye Mountain, which are less affected by wind and offer better habitats. The high ecologically sensitive areas are mainly the elevated regions of certain mountains, some lower-elevation areas of windbreaks along the seashore, encompassing more windward areas, and some mid-slope leeward areas. The ecologically insensitive areas mainly comprise urban built-up areas and development zones in the central and southern parts of the island, along with rural settlements in the central and northern regions. The extreme ecologically sensitive areas are primarily located on the windward slopes and ridges of Jun Mountain, Wangye Mountain, and Niuzhai Mountain at an elevation of about 200 m.

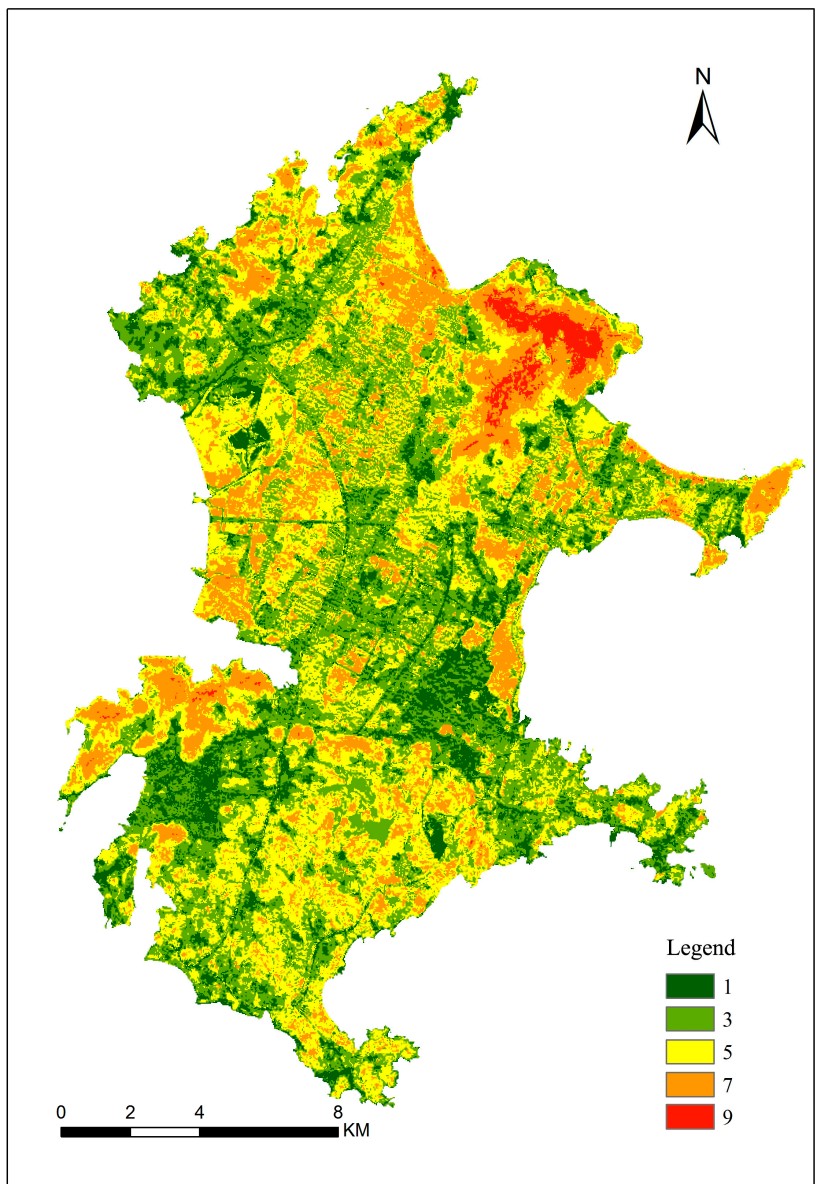

**Figure 5.** Comprehensive ecological sensitivity. Note: 1 represents ecological insensitivity; 3 represents slight ecological sensitivity; 5 represents moderate ecological sensitivity; 7 represents high ecological sensitivity; 9 represents extreme ecological sensitivity.

*3.2. Analysis of Montane Plant Communities in Various Ecologically Sensitive Areas of Pingtan Island*

3.2.1. Analysis of the Species Composition of Montane Plant Communities in Various Ecologically Sensitive Areas

A total of 93 species (82 genera and 47 families) of mountain plants were investigated in 33 sample sites. Among the 93 species, 13 were trees, 35 were shrubs, and 45 were herbs. There are 2 families with 6–9 species (17.28%), 13 families with 2–5 species (53.09%), and 24 families with 1 species (29.63%). Among them, 59 species (54 genera and 38 families) were found in moderate ecologically sensitive areas, 81 species (73 genera and 45 families) in high sensitive areas, and 33 species (32 genera and 30 families) in extreme ecologically sensitive areas (Table A1).

### 3.2.2. Analysis of Dominant Species of Montane Plant Communities in Various Ecologically Sensitive Areas

A total of 13 *A. confusa* forests, 4 *C. equisetifolia* forests, 6 *P. thunbergii* forests, 2 *P. elliottii* forests, 1 *Schima superba* forests, 3 *A. confusa* scrubs, 3 *E. emarginata* scrubs, and 1 *P. thunbergii* scrub were identified in the 33 sample plots (Table A2).

In the moderate ecologically sensitive area, J2, J3, N2, N3, and W3 sample plots all have *A. confusa* forests as constructive species, J6 is a pure forest of C. equisetifolia, J8 is a forest of *S. superba* mixed with *A. confusa* and *Cunninghamia lanceolata*, and J16 is a forest of *P. thunbergii*. Among the high ecologically sensitive areas, the group species in plots J1, J15, N1, W1, W2, W5, W6, and W7 are all *A. confusa* forests; J13, N4, and N5 are *P. thunbergii* forests; J7 and J9 are pure and mixed *P. elliottii* forests, respectively; and J4, J5, and N7 are pure forests of *C. equisetifolia*. The rest of the sample plots are missing the tree layer: J14, N6, and W10 are *A. confusa* scrub, W4 is *P. thunbergii* scrub, W8 and W9 are *E. emarginata* scrub, and the various plant species in the scrubs are auxiliary species to each other. In the extreme ecologically sensitive area, J10 and J11 are *P. thunbergii* forests, and J12 is *E. emarginata* scrub.

### 3.2.3. Characterization of Montane Plant Communities in Various Ecologically Sensitive Areas
#### Characterization of Plant Communities

The characterization of the various sensitive areas revealed that the canopy density (CD) and mean height of tree layer (MHTL) exhibited a pattern of moderate ecologically sensitive areas > high ecologically sensitive areas > extreme ecologically sensitive areas. The mean diameter at breast height (MDBH) and the mean height of shrub layer (MHSL) have a pattern of high ecologically sensitive areas > moderate ecologically sensitive areas > extreme ecologically sensitive areas. The mean height of herb layer (MHHL) was the highest in the extreme ecologically sensitive area (Table 4).

**Table 4.** Means and standard deviations of characteristics of plant communities in various ecologically sensitive areas.

| Ecologically Sensitive Area | CD | MDBH (cm) | MHTL (m) | MHSL (cm) | MHHL (cm) |
|---|---|---|---|---|---|
| Moderate ecologically sensitive areas | $74.00 \pm 30.41$ | $8.17 \pm 3.15$ | $4.98 \pm 1.84$ | $0.61 \pm 0.38$ | $0.29 \pm 0.08$ |
| High ecologically sensitive areas | $49.59 \pm 42.61$ | $8.42 \pm 6.99$ | $4.72 \pm 3.89$ | $0.71 \pm 0.32$ | $0.27 \pm 0.15$ |
| Extreme ecologically sensitive areas | $33.33 \pm 35.12$ | $4.66 \pm 4.04$ | $1.99 \pm 1.77$ | $0.59 \pm 0.05$ | $0.29 \pm 0.26$ |

Note: CD represents canopy density; MDBH represents mean diameter at breast height; MHTL represents mean height of tree layer; MHSL represents mean height of shrub layer; MHHL represents mean height of herb layer.

Concerning different plant communities, the tree layer exhibited better growth conditions in *A. confusa* and *C. equisetifolia* forests, while *P. thunbergii* forests demonstrated the lowest performance in CD, MDBH, and MHTL. Shrub layer and grass layer growth heights were optimal in *E. emarginata* scrub, while shrub layer growth was minimal in *C. equisetifolia* forests, and herb layer growth was lowest in *P. elliottii* forests (Table 5).

**Table 5.** Means and standard deviations of characteristics of various plant communities.

| Plant Community | CD | MDBH (cm) | MHTL (m) | MHSL (cm) | MHHL (cm) |
|---|---|---|---|---|---|
| *A. confusa* forest | $85.77 \pm 5.09$ | $9.31 \pm 1.99$ | $5.61 \pm 0.91$ | $0.75 \pm 0.34$ | $0.29 \pm 0.15$ |
| *C. equisetifolia* forest | $82.0 \pm 8.71$ | $17.86 \pm 6.40$ | $10.10 \pm 1.87$ | $0.33 \pm 0.18$ | $0.21 \pm 0.18$ |
| *P. thunbergia* forest | $16.67 \pm 28.75$ | $6.44 \pm 1.80$ | $2.48 \pm 0.63$ | $0.62 \pm 0.15$ | $0.27 \pm 0.12$ |
| *P. elliottii* forest | $74.0 \pm 5.66$ | $13.55 \pm 6.46$ | $8.47 \pm 2.75$ | $0.56 \pm 0.06$ | $0.16 \pm 0.23$ |
| *A. confuse* scrub | 0 | 0 | 0 | $0.55 \pm 0.12$ | $0.33 \pm 0.05$ |
| *E. emarginata* scrub | 0 | 0 | 0 | $1.08 \pm 0.49$ | $0.41 \pm 0.19$ |

Note: CD represents canopy density; MDBH represents mean diameter at breast height; MHTL represents mean height of tree layer; MHSL represents mean height of shrub layer; MHHL represents mean height of herb layer.

In moderate ecologically sensitive areas, the CD, MDBH, and MHTL were higher for *C. equisetifolia* forests and the lowest for *P. thunbergii* forests. The shrub layer height was the highest in *A. confusa* forests and the lowest in *C. equisetifolia* forests. The herb layer was taller in *C. equisetifolia* forests. In high ecologically sensitive areas, *A. confusa* and *C. equisetifolia* forests had high levels of CD, and *C. equisetifolia* and *P. elliottii* forests had higher MDBH and MHTL. The shrub layer height reached its maximum in *E. emarginata* scrub, the herb layer was taller in *P. thunbergii* forests, and both the shrub and herb layers were at their minimum in *C. equisetifolia* forests. In the extreme ecologically sensitive areas, the underlying characteristic data were highest for *P. thunbergii* forests, except for the herb layer height. Additionally, shrub layer heights were similar (Figure 6).

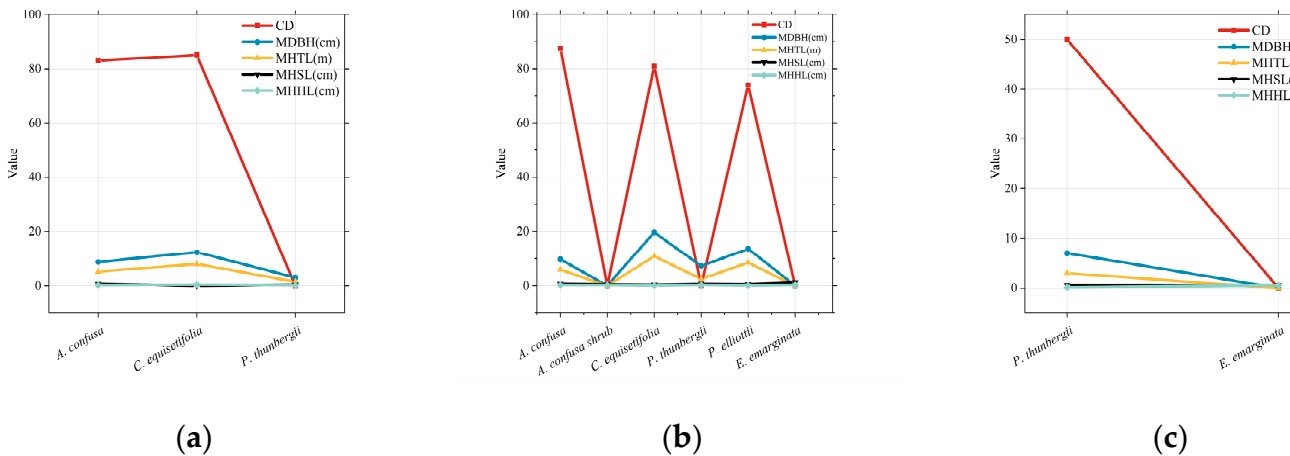

**Figure 6.** Plant community characteristics in various ecologically sensitive areas. (**a**) Moderate ecological sensitivity, (**b**) High ecological sensitivity, (**c**) Extreme ecological sensitivity. Note: CD represents canopy density; MDBH represents mean diameter at breast height; MHTL represents mean height of tree layer; MHSL represents mean height of shrub layer; MHHL represents mean height of herb layer.

Species Diversity

As depicted in Table 6, the Shannon–Wiener index, Pielou index, Margalef index, and Simpson index in the three sensitive areas exhibited an upward trend with increasing ecological sensitivity.

**Table 6.** Means and standard deviations of plant community diversity in various ecologically sensitive areas.

| Ecologically Sensitive Area | Shannon–Wiener | Pielou | Margalef | Simpson |
| --- | --- | --- | --- | --- |
| Moderate ecologically sensitive areas | 2.25 ± 0.50 | 0.82 ± 0.12 | 3.13 ± 0.91 | 0.83 ± 0.12 |
| High ecologically sensitive areas | 2.27 ± 0.47 | 0.84 ± 0.08 | 3.24 ± 1.08 | 0.86 ± 0.08 |
| Extreme ecologically sensitive areas | 2.51 ± 0.22 | 0.89 ± 0.03 | 3.33 ± 0.85 | 0.90 ± 0.02 |

Among the various plant communities, all four diversity indices for *A. confusa* forests were the highest, with scrub diversity surpassing that of forests. Subsequently, *E. emarginata* scrub exhibited higher Shannon–Wiener and Simpson indices, while *P. thunbergii* forests showed a higher Margalef index. In contrast, the combined species diversity of *C. equisetifolia* and *P. elliottii* forests was comparatively low (Table 7).

**Table 7.** Means and standard deviations of species diversity of various plant communities.

| Plant Community | Shannon–Wiener | Pielou | Margalef | Simpson |
|---|---|---|---|---|
| *A. confusa* forest | 2.42 ± 0.07 | 0.87 ± 0.01 | 3.38 ± 0.18 | 0.88 ± 0.01 |
| *C. equisetifolia* forest | 1.64 ± 0.32 | 0.86 ± 0.05 | 1.95 ± 0.81 | 0.75 ± 0.07 |
| *P. thunbergia* forest | 2.38 ± 0.12 | 0.83 ± 0.05 | 3.50 ± 0.20 | 0.89 ± 0.01 |
| *P. elliottii* forest | 1.72 ± 0.34 | 0.79 ± 0.08 | 2.10 ± 0.57 | 0.77 ± 0.09 |
| *A. confuse* scrub | 2.69 ± 0.23 | 0.86 ± 0.04 | 4.39 ± 0.58 | 0.91 ± 0.02 |
| *E. emarginata* scrub | 2.45 ± 0.07 | 0.85 ± 0.01 | 3.13 ± 0.38 | 0.89 ± 0.00 |

Among the moderate ecologically sensitive areas, the Margalef index exhibited higher values in *A. confusa* forests, while the Shannon–Wiener, Pielou, and Simpson indices were comparable between *P. thunbergii* and *A. confusa*, and all were lowest in *C. equisetifolia* forests. Within the high ecologically sensitive areas, the Shannon–Wiener, Margalef, and Simpson indices reached their peaks in *A. confusa* scrub and were lower in both *P. elliottii* and *C. equisetifolia* forests. The Pielou index recorded its highest value in *C. equisetifolia* forests, followed by *A. confusa* forests and scrubs, and was at its lowest in *P. thunbergii* forests. Among extreme ecologically sensitive areas, all four diversity indices attained their highest values in *P. thunbergii* forests (Figure 7).

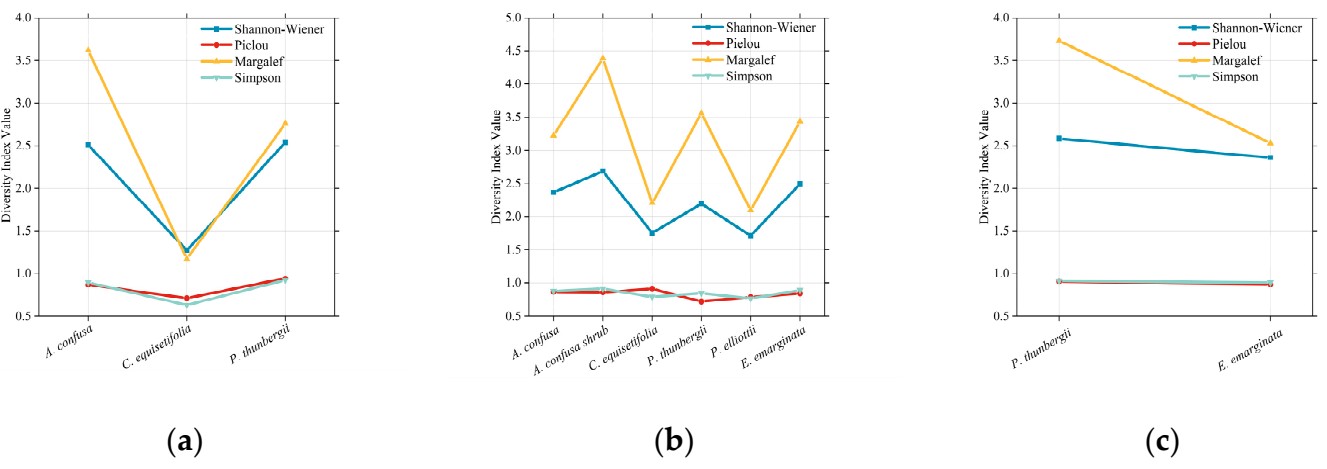

(**a**)        (**b**)        (**c**)

**Figure 7.** Diversity indices of plant community in various ecologically sensitive areas. (**a**) Moderate ecological sensitivity, (**b**) High ecological sensitivity, (**c**) Extreme ecological sensitivity.

## 4. Discussion

### 4.1. Ecological Sensitivity Characteristics of Pingtan Island

The results of the ecological sensitivity evaluation of Pingtan Island reveal that the elevation, slope, soil, and water on Pingtan Island exhibit a low level of ecological sensitivity, while vegetation coverage demonstrates high ecological sensitivity. The wind environment is categorized as medium-low ecological sensitivity and the land use type as medium-high ecological sensitivity, and the distribution of the slope direction is relatively uniform. The results of the single-factor weight analysis indicate that the soil, wind environment, and land use type are the more influential factors, aligning with previous research findings that attribute the vulnerability of Pingtan Island's ecological environment to a combination of natural conditions and human activities [17]. It is recommended that ecological protection and improvement efforts on Pingtan Island prioritize the enhancement of soil conditions, the wind environment, and the optimization of land use types. Specifically, the spatial distribution analysis of soil sensitivities on Pingtan Island highlights severe salinization and sanding in the four vents, necessitating targeted prevention measures. In the subsequent stages of plant landscape construction, the ecological protection role of windbreaks should be reinforced. Considering the prevailing northeasterly winds on Pingtan Island, due

north and northeasterly slopes emerge as the primary windward areas. Among them, the Changjiang'Ao windbreak, where the east and west of the stream have the maximum wind frequency and the dominant wind flow through them, is the location of the concentrated distribution of perennial wind resources. This area should be judiciously utilized for wind energy and plant configuration planning. Conversely, lower wind speeds are observed in the southern and western parts of the island, most of the southern part of Jun Mountain, the northern mountainous areas, and some of the lower-elevation areas, indicating that major mountainous terrains play a significant role in reducing wind speeds [37]. Water is the most sensitive land use type on Pingtan Island due to topographical constraints, as is common for many islands globally, given their limited freshwater resources [2]. Despite this limitation, water is indispensable to the ecosystem. Woodland ranks as the second most sensitive type due to the inherent difficulty in restoring mountains once destroyed. Moreover, woodlands contribute significantly to ecosystem regulation and supporting services. This is particularly crucial for islands, where plants with specialized functions such as wind protection and sand fixation hold heightened importance.

From the perspective of comprehensive ecological sensitivity, Pingtan Island is dominated by three categories: slightly ecologically sensitive areas (35.72%), moderate ecologically sensitive areas (33.99%), and extreme ecologically sensitive areas (18.02%). The ecologically insensitive areas are primarily situated in regions with notable human activity, featuring low elevation, flat terrain, sparse vegetation, and robust resistance to disturbances. These areas permit relatively intense urban development and construction and are also the best areas for the diversification of plant landscape construction. The slightly ecologically sensitive areas encompass some mountainous and green regions where human activities introduce certain disturbances. Nevertheless, the ecosystems in these areas are more stable than those in insensitive regions. Consequently, a combination of plant and landscape development with human activities conducted in a moderate manner is feasible. The moderate ecologically sensitive areas include sporadic rural settlements with less human interference, featuring higher vegetation density compared to areas with lower sensitivity. Plant landscapes in these zones can be adjusted accordingly, with options such as forest phase modifications to enhance landscape effects. Additionally, localized forest recreational activities can be conducted in areas with suitable microclimatic environments. The high ecologically sensitive areas boast high ecological quality and are abundant in botanical landscape resources. These areas warrant improvements grounded in ecological protection and the enhancement of ecological landscapes. The vegetation in extreme ecologically sensitive areas experiences minimal human disturbance but is more susceptible to natural environmental factors such as wind and sun, leading to the stunted growth and desiccation of vegetation. These zones are primarily designated for ecological restoration and protection, with the goal of constructing a distinctive ecological landscape on the island.

The wind environment simulation in this study was conducted by simulating island-wide near-surface wind conditions of NNE 13.9 m/s (gale force 7) without accounting for the topography of plants and buildings. However, upon comparing these simulation results with field measurement data, a notable similarity is observed, affirming the feasibility of the wind environment simulation method employed in this study. This finding is intended to provide methodological support for future related studies. In practical applications, the presence of wooded areas and buildings across more extensive parts of the island, along with the potential for improved localized microclimatic environments in various ecologically sensitive areas due to ground plants and shading by buildings, necessitates further in-depth investigations into these localized microclimatic habitats. Future efforts should focus on refining both the model and observation conditions to enhance the overall reliability of the conclusions drawn. Given the variations in habitat characteristics among islands, those with limited conditions can consider referring to this model for the study of island habitats.

*4.2. Typical Plant Communities and Characteristics in the Mountains of Pingtan Island across Various Ecologically Sensitive Areas*

(1) Exploration of Composition and Dominant Species of Typical Mountain Plant Community

The survey revealed that the typical plant community in the mountains of Pingtan Island follows the order herb layer > shrub layer > tree layer, aligning with the stage dominated by the shrub–grass succession stage. This pattern is consistent with previous findings that herbs are generally considered to be more adept at colonization than woody plants [38]. This phenomenon can be ascribed to the impact of strong winds, rendering low-growing herbaceous plants more adaptable to this environment in comparison to taller trees and shrubs [21]. Soil conditions, climate, and other factors also contribute to these observed patterns. Among the various ecologically sensitive areas, the high ecologically sensitive areas exhibit characteristics such as a greater number of species and higher species diversity. However, the overall mountainous plant community on Pingtan Island remains relatively homogeneous, featuring a significant proportion of pure forests. There exists a certain degree of repetition and similarity in the plant communities across different ecologically sensitive areas. This phenomenon can be attributed to the unique habitat conditions of islands, where natural selection favors more adaptable native plant communities. Consequently, the plant community similarity is more pronounced.

The dominant species vary across different sensitive areas. The group species in moderate ecologically sensitive areas are all forests, dominated by *A. confusa* and *S. superba* forests. The high and extreme ecologically sensitive areas comprise both arborvitae forests and scrub. The high ecologically sensitive areas are dominated by *A. confusa*, *P. thunbergii*, *P. elliottii*, and *C. equisetifolia* forests, while the extreme ecologically sensitive areas are dominated by *P. thunbergii* forests and *E. emarginata* scrub. Observing the different ecologically sensitive areas, it is evident that with increasing elevation, there is a shift in dominance from *A. confusa* to *P. thunbergii* and finally to *E. emarginata*, indicating a successional pattern.

In general, the current montane plant community on Pingtan Island is less diverse, with a single structure and lower species diversity. This is due to the fact that, because of the need to adapt to habitats with persistent high winds and low soil nutrients, Pingtan Island has more wild herbaceous plants, and more plants tend to have smaller flowers, leaves, and fruits [21,39], with small and hairy, highly lignified, and fleshy leaves and more thorny shrubs and vines [40]. In the future, we should continue to screen suitable plants for mountainous areas with suitable local characteristics, strengthen the adaptability of plants to windy environments and infertile soils through domestication and introduction in order to increase the diversity of mountainous plants, and further strengthen the research on typical plant patterns in mountainous areas so as to enhance the significance of practical guidance.

(2) Characterization of Typical Mountain Plants

There are variations in the community characteristics of plant communities across different ecologically sensitive areas, which are characterized by the dominance of differentiated tree–shrub–herb layers. Trees exhibit superior growth in moderate ecologically sensitive areas, while high ecologically sensitive areas transition toward a tree–shrub community. Herbs grow better in extreme ecologically sensitive areas compared to other areas. These differences are associated with the habitat conditions in distinct sensitive areas, wherein shrubs, grasses, and other lower plants show better adaptation to the environment in higher-elevation areas [41].

The analysis of the different plant communities revealed that the tree growth of *C. equisetifoli* forests was more robust in both the moderate and high ecologically sensitive areas. However, the prominence of this dominant species has led to an excessive canopy density, creating unfavorable growing conditions for other smaller trees or lower plants. Consequently, shrub growth is minimal, and herbaceous growth is suboptimal. In the case of the *P. thunbergii* forest, its performance is moderate in both the moderate and high ecologically sensitive areas. It exhibits the lowest level of tree canopy density, but this

characteristic facilitates the provision of necessary light and other conditions for the growth of other plants. Therefore, shrubs and grasses within this forest exhibit better growth. The *E. emarginata* scrub predominantly occurs in high and extreme ecologically sensitive areas, where the absence of the tree layer allows shrubs and grasses to thrive due to increased sunlight availability and more favorable growing conditions. The findings from the species diversity study indicate that increased ecological sensitivity contributes to the species diversity of plant communities. However, the disparities in their species diversity were not statistically significant. This is attributed to the influence of environmental factors, including elevation, precipitation, wind speed, and soil, on island species diversity [42–45]. In the study area, the overall habitats do not vary significantly, which subsequently leads to small differences in overall species diversity.

Species diversity indicators exhibited more pronounced variations among plant communities. The Shannon–Wiener, Pielou, Margalef, and Simpson indices within *A. confusa* plant communities, particularly in *A. confus* scrub, are elevated. This phenomenon may be attributed to the interspersion of scrub with young trees. Young trees in the community increase the diversity of the shrub layer, and the plants are more light-adapted [29,46] The moderate and high ecologically sensitive areas exhibited a heightened degree of diversity in the context of *A. confusa*. This phenomenon may be attributed to the successional stage of the *A. confusa* community on the middle and lower slopes, encompassing both sun- and shade-loving plants. Furthermore, certain anthropogenic disturbances have created forest gaps in *A. confusa* forests, where the understory of shrubby *S. superba* and *Ficus formosana* receives increased light exposure, gaining a competitive advantage and developing into trees. This dynamic contributes to an augmentation in the diversity of this community. Additionally, the leeward slope location of *A. confusa* forests favors the growth of various plant types, as they are less affected by high winds. Pielou and Simpson indices were also high in *P. thunbergii* forests, indicating structural stability within the plant community [47], a characteristic linked to community succession patterns. The relatively low canopy density in *P. thunbergii* forests plays a role in supporting the growth of other sun-loving plants. The low species diversity of *C. equisetifolia* forests stems from a lack of diversity or very low diversity in the arboreal shrub layer. Although individual plant growth is robust, the extremely high canopy density creates unfavorable conditions for the growth of other sun-loving plants or lower scrubs, resulting in low species diversity. Additionally, the reduction in plant numbers can impede the provision of sufficient apoplastic material for microbial decomposition, subsequently leading to habitat deterioration and a decline in species diversity indices [29]. As ecological sensitivity increases, it signifies more pronounced ecological challenges. Based on the distribution patterns, it is observed that *P. thunbergii* forests on Pingtan Island are predominantly situated in uphill positions and northeast slope directions, implying their preference for more windward locations. This suggests that *P. thunbergii* forests exhibit enhanced adaptability to windy conditions and the overall environment. On the other hand, *P. thunbergii* forests also offer some shaded spaces, facilitating growth conditions for semi-sun-loving plants like *Melastoma candidum* and subsequently enhancing their overall diversity.

In terms of the amalgamation of characteristics and species diversity, the ideal ecological communities that are more stable in the mountains of Pingtan Island include *A. confusa* forests and scrubs, *E. emarginata* scrubs, *P. thunbergii* forests, etc. Conversely, those with lower composite characteristic indices were predominantly *C. equisetifolia* forests, *P. elliottii* forest communities, etc., consisting of pure forests with diminished species diversity and consequently exhibiting lower stability. Based on these findings, landscape construction in different sensitive areas can be implemented by referencing the corresponding plant community model, thereby providing a model reference for the landscape construction of other analogous islands and mountains.

## 5. Conclusions

The unique geographical location and environmental characteristics of islands have led to fragile ecosystems and distinctive features in plant communities. Therefore, it is crucial to delineate the characteristics of plant communities in various ecologically sensitive areas for the protection and sustainable development of island ecosystems. This paper focuses on Pingtan Island in Fujian Province, China, and draws the following conclusions: (1) A proficient approach to wind environment modeling and data acquisition for Pingtan Island is proposed. The outcomes of the ecological sensitivity evaluation conducted on this basis revealed that Pingtan Island exhibits an ecological sensitivity distribution pattern of slightly ecologically sensitive areas > moderate ecologically sensitive areas > high ecologically sensitive areas > ecologically insensitive areas > extreme ecologically sensitive areas. Key influencing factors include the soil texture (0.193), wind environment (0.162), and land use type (0.143). Accordingly, this paper proposes corresponding strategies for ecological environmental protection tailored to different ecologically sensitive areas. (2) The mountainous habitats of Pingtan Island are distributed in moderate, high, and extreme ecologically sensitive areas. This paper summarizes the characteristics of more stable montane plant communities within different ecologically sensitive areas through investigation and analysis. Overall, following the pattern of plant succession, herbs are the most abundant, followed by shrubs and, finally, trees. The number of plants increases with sensitivity. The base characteristics of trees are better in the moderate ecologically sensitive areas, and the high ecologically sensitive areas exhibit a trend toward tree–shrub communities. Among these two types of ecologically sensitive areas, *A. confusa* forests show the best overall ecological benefits in tree forests. *C. equisetifolia* forests, despite having superior tree-based characteristics, display lower species diversity, whereas *P. thunbergii* forests exhibit the opposite pattern. Additionally, the combined ecological benefits of *A. confusa* and *E. emarginata* scrubs are superior. Herbs thrive in extreme ecologically sensitive areas, showing the best combined ecological benefits with *P. thunbergii* forests.

This study encapsulates the ecological sensitivity characteristics of Pingtan Island as well as the characteristics of plant communities suitable for different ecologically sensitive areas and provides corresponding optimization strategies. The objective is to systematically undertake targeted initiatives for island plant restoration and ecosystem preservation, thereby fostering the sustainable development of the island's habitats.

**Author Contributions:** Conceptualization, J.L. (Jinyan Liu), J.L. (Junyi Li), G.D. and J.D.; methodology, J.L. (Jinyan Liu) and J.L. (Junyi Li); software, J.L. (Jinyan Liu), J.L. (Junyi Li) and D.C.; validation, J.L. (Jinyan Liu), J.L. (Junyi Li) and D.C.; formal analysis, J.L. (Jinyan Liu), J.L. (Junyi Li) and L.G.; investigation, J.L. (Jinyan Liu), J.L. (Junyi Li), D.C. and L.G.; resources, J.L. (Jinyan Liu), J.L. (Junyi Li), G.D. and J.D.; data curation, J.L. (Jinyan Liu), J.L. (Junyi Li) and L.G.; writing—original draft preparation, J.L. (Jinyan Liu) and J.L. (Junyi Li); writing—review and editing, J.L. (Jinyan Liu), J.L. (Junyi Li), G.D. and J.D.; visualization, J.L. (Jinyan Liu), J.L. (Junyi Li), D.C. and L.G.; supervision, J.L. (Jinyan Liu), J.L. (Junyi Li), G.D. and J.D.; project administration, G.D. and J.D.; funding acquisition, G.D. and J.D. All authors have read and agreed to the published version of the manuscript.

**Funding:** This research was funded by Green Urbanization across China and Europe: Collaborative Research on Key Technological Advances in Urban Forests, grant number 2021YFE0193200.

**Institutional Review Board Statement:** Not applicable.

**Informed Consent Statement:** Not applicable.

**Data Availability Statement:** The data used to support the findings of this study are available from the corresponding author upon request.

**Conflicts of Interest:** The authors declare no conflicts of interest.

## Appendix A

**Table A1.** Species composition of a typical sample site in the mountains of Pingtan Island.

| Sensitivity Zones | Number of Species (Families) | Family Name (Genus: Species) | Percentage (Family, Genus, Species) |
|---|---|---|---|
| Moderate ecologically sensitive area | 2–5 (11) | Labiatae (2:2), Pteridaceae (3:3), Gramineae (4:4), Apocynaceae (3:3), Thelypteridaceae (1:3), Malvaceae (2:2), Compositae (3:4), Rubiaceae (3:3), Rosaceae (2:4), Salicaceae (2:2), Umbelliferae (2:2) | 29%, 50%, 54% |
| | 1 (27) | Myrtaceae, Smilacaceae, Pentaphylacaceae, Lauraceae, Sapindaceae, Euphorbiaceae, Lygodiaceae, Pittosporaceae, Pinaceae Lindl., Asparagaceae, Caprifoliaceae, Thymelaeaceae, Rutaceae, Gleicheniaceae, Theaceae, Casuarinaceae, Cannabaceae, Menispermaceae, Celastraceae, Rhamnaceae, Asphodelaceae, Cupressaceae Bartling, Vitaceae, Moraceae, Leguminosae, Melastomataceae, Primulaceae | 71%, 50%, 46% |
| | 6–9 (2) | Gramineae (6:6), Compositae (7:7) | 4.4%, 18%, 16% |
| High ecologically sensitive area | 2–5 (13) | Labiatae (2:2), Leguminosae (4:4), Pteridaceae (3:5), Apocynaceae (3:3), Thelypteridaceae (1:3), Malvaceae (2:2), Rubiaceae (3:3), Rosaceae (2:3), Moraceae (2:3), Pinaceae Lindl. (1:2), Asparagaceae (3:3), Salicaceae (2:3), Umbelliferae (2:2) | 28.9%, 41%, 47% |
| | 1 (30) | Smilacaceae, Araliaceae, Pentaphylacaceae, Goodeniaceae, Lauraceae, Sapindaceae, Euphorbiaceae, Lygodiaceae, Pittosporaceae, Elaeagnaceae, Polygonaceae, Umbelliferae, Caprifoliaceae, Thymelaeaceae, Rutaceae, Gleicheniaceae, Theaceae, Casuarinaceae, Menispermaceae, Celastraceae, Rhamnaceae, Asphodelaceae, Cupressaceae Bartling, Phytolaccaceae, Vitaceae, Nephrolepidaceae, Myrtaceae, Lindsaeaceae, Melastomataceae, Primulaceae | 66.7%, 41%, 37% |
| Extreme ecologically sensitive area | 2–5 (3) | Gramineae (2:2), Rubiaceae (2:2), Rosaceae (1:2) | 10%, 16%, 18% |
| | 1 (27) | Asphodelaceae, Smilacaceae, Primulaceae, Labiatae, Cannabaceae, Leguminosae, Menispermaceae, Pteridaceae, Lygodiaceae, Elaeagnaceae, Apocynaceae, Malvaceae, Compositae, Gleicheniaceae, Vitaceae, Solanaceae, Thymelaeaceae, Umbelliferae, Moraceae, Pinaceae Lindl., Asparagaceae, Celastraceae, Pentaphylacaceae, Salicaceae, Melastomataceae, Rutaceae, Lauraceae | 90%, 84%, 82% |

**Table A2.** Dominant species in plant communities in typical sample sites in the mountains of Pingtan Island.

| Sensitivity Zones | No. | Name of Plant Community | Plant Community |
|---|---|---|---|
| Moderate ecologically sensitive area | J2 | *A. confuse + Litsea rotundifolia—Zanthoxylum nitidum + L. rotundifolia—M. floridulus cluster* | forest |
| | J3 | *A. confusa + Celtis sinensis—L. rotundifolia—Oplismenus undulatifolius cluster* | forest |
| | J6 | *C. equisetifolia—Z. nitidum—O. undulatifolius cluster* | forest |
| | J8 | *S. superba + Cunninghamia lanceolata—S. superba-Adiantum capillus-veneris cluster* | forest |
| | J16 | *P. thunbergii-Dodonaea viscosa—M. floridulus cluster* | forest |
| | N2 | *A. confusa + Casearia glomerata—C. glomerata—Lygodium japonicum cluster* | forest |
| | N3 | *A. confusa +P. thunbergii—M. candidum—Cyclosorus acuminatus cluster* | forest |
| | W3 | *A. confusa—Nerium indicum—Dicranopteris dichotoma cluster* | forest |

**Table A2.** *Cont.*

| Sensitivity Zones | No. | Name of Plant Community | Plant Community |
|---|---|---|---|
| High ecologically sensitive area | J1 | *A. confusa + L. rotundifolia—Ardisia crenata-Nephrolepis auriculata cluster* | forest |
| | J4 | *C. equisetifolia—C. glomerata—O. undulatifolius cluster* | forest |
| | J5 | *C. equisetifolia—L. rotundifolia—Psychotria serpens cluster* | forest |
| | J7 | *P. elliottii—M. candidum—Dianella ensifolia cluster* | forest |
| | J9 | *P. elliottii + C. equisetifolia—C. equisetifolia + L. rotundifolia—D. ensifolia cluster* | forest |
| | J13 | *P. thunbergii—E. emarginata + A. confusa—M. floridulus cluster* | forest |
| | J14 | *A. confusa + M. candidum—Setaria viridis cluster* | scrub |
| | J15 | *A. confusa + L. rotundifolia—L. rotundifolia + A. crenata—D. dichotoma cluster* | forest |
| | N1 | *A. confusa + C. glomerata—C. glomerata-D. dichotoma cluster* | forest |
| | N5 | *P. thunbergii + E. emarginata—D. viscosa—Ischaemum indicum cluster* | forest |
| | N6 | *A. confusa + E. emarginata—D. ensifolia cluster* | scrub |
| | N7 | *C. equisetifolia—E. emarginata—M. floridulus cluster* | forest |
| | W1 | *A. confusa—Xylosma racemosum—D. ensifolia cluster* | forest |
| | W2 | *A. confusa + F. formosana—E. emarginata—A. capillus-veneris cluster* | forest |
| | W3 | *A. confusa—N. indicum—D. dichotoma cluster* | forest |
| | W4 | *P. thunbergia + A. confusa—Artemisia capillaris cluster* | scrub |
| | W5 | *A. confusa + S. superba—Cudrania cochinchinensis—D. ensifolia cluster* | forest |
| | W6 | *A. confusa + S. superba—F. formosana—A. capillus-veneris cluster* | forest |
| | W7 | *A. confusa + F. formosana—C. cochinchinensis—C. acuminatus cluster* | forest |
| | W8 | *E. emarginata + M. candidum—D. ensifolia cluster* | scrub |
| | W9 | *E. emarginata + A. confusa—I. indicum cluster* | scrub |
| | W10 | *A. confusa + E. emarginata—D. ensifolia cluster* | scrub |
| Extreme ecologically sensitive area | J10 | *P. thunbergii + C. sinensis—E. emarginata—O. undulatifolius cluster* | forest |
| | J11 | *P. thunbergii—E. emarginata + M. candidum—D. dichotoma cluster* | forest |
| | J12 | *E. emarginata + A. confusa—M. floridulus cluster* | scrub |

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
