# Peer review of "Differential Analysis of Island Mountain Plant Community Characteristics: Ecological Sensitivity Perspectives"

_sustainability, doi:10.3390/su16051988_

Round 1

Reviewer 1 Report

Comments and Suggestions for Authors

The manuscript is very well written the idea of the manuscript is very good. I have no major comments on the manuscript. I have just wished the authors gave a brief on the plant communities history and the ecological succession in the three different locations (e.g. native plant community, the introduced plant species, the invasive species) somewhere in the introduction.

Reviewer 2 Report

Comments and Suggestions for Authors

There is a noticeable lack of knowledge of the use of professional terms in the field of plant ecology. Some terms are incorrect or used incorrectly. In some places in the text, the descriptions of plant communities are written subjectively and not enough with appropriate scientific terms. In the end, the discussion and conclusions do not clearly state what the research should bring: the protection of habitat types in the investigated locations or something else?

Reviewer 3 Report

Comments and Suggestions for Authors

This study focuses on using 8 indicators of Ecological Sensitivity, including elevation, slope direction, plant cover, soil texture, water environment, wind environment, and land use type. I have some comments on the manuscript as follow:

1- Keywords, please do not repeat any word already mentioned in the title

2- References (mainly in the section of Introduction), several of these refs. are old and need to be updated, publication in 2024, 2023, 2022 should be the most cited as the authors can added.

3- Section of Introduction: Why didn’t the authors explain the role of the studied ecological indicators?

4- Materials and methods: Where are the soil analyses? Refs.? Because the color or texture of soil is not enough.

5- Topography and Geomorphology, topography was explained but the geomorphology, nothing.

6- Flowchart is needed to explain this study, please

7- About the soil texture, “Brick Red Loamy Soil, or Reddish Loamy Soil” use the international soil texture classes, please

8- The following line 257 needs a ref. “Plant communities were classified based on plant community-ecological principles. “

9- In general, many parts in section of Materials and Methods need a ref.

10- Please replace “2.3.3. Data Analysis” by 2.3.3. Geostatistical Analyses

11- Table 4and any table or Figure must explain any abbreviation without referring to the text, please

12- Figures 5 and 6 are very poor quality, should be improved

13- Where are the indicators of land degradation in this study as mentioned in the text?

14- Why didn’t the authors study both: Land Suitability and Capability?

Round 2

Reviewer 3 Report

Comments and Suggestions for Authors

Thanks to the authors for responding to all comments in a good manner. I will accept the manuscript in this form